# Mechanism of Albuminuria Reduction by Chymase Inhibition in Diabetic Mice

**DOI:** 10.3390/ijms21207495

**Published:** 2020-10-11

**Authors:** Kentaro Terai, Denan Jin, Kenji Watase, Akihisa Imagawa, Shinji Takai

**Affiliations:** 1Department of Innovative Medicine, Graduate School of Medicine, Osaka Medical College, 2-7 Daigaku-machi, Takatsuki, Osaka 569-8686, Japan; in1381@osaka-med.ac.jp (K.T.); pha012@osaka-med.ac.jp (D.J.); kenzzy36@yahoo.co.jp (K.W.); 2Department of Internal Medicine (I), Osaka Medical College, 2-7 Daigaku-machi, Takatsuki, Osaka 569-8686, Japan; imagawa@osaka-med.ac.jp

**Keywords:** chymase, inhibitor, diabetes, kidney, albuminuria

## Abstract

Chymase has several functions, such as angiotensin II formation, which can promote diabetic kidney disease (DKD). In this study, we evaluated the effect of the chymase inhibitor TY-51469 on DKD in diabetic db/db mice. Diabetic mice were administered TY-51469 (10 mg/kg/day) or placebo for 4 weeks. No significant difference was observed in body weight and fasting blood glucose between TY-51469- and placebo-treated groups. However, a significant reduction in urinary albumin/creatinine ratio was observed in the TY-51469-treated group compared with the placebo-treated group. In the renal extract, chymase activity was significantly higher in placebo-treated mice than in non-diabetic db/m mice, but it was reduced by treatment with TY-51469. Both NADPH oxidase 4 expression and the oxidative stress marker malondialdehyde were significantly augmented in the placebo-treated group, but they were attenuated in the TY-51469-treated group. Significant increases of tumor necrosis factor-α and transforming growth factor-β mRNA levels in the placebo-treated group were significantly reduced by treatment with TY-51469. Furthermore, the expression of nephrin, which is a podocyte-specific protein, was significantly reduced in the placebo-treated group, but it was restored in the TY-51469-treated group. These findings demonstrated that chymase inhibition reduced albuminuria via attenuation of podocyte injury by oxidative stress.

## 1. Introduction

Diabetic kidney disease (DKD) is the principal cause of renal failure worldwide. Angiotensin II promotes the progression of DKD, and angiotensin-converting enzyme (ACE) inhibitors and angiotensin II receptor blockers (ARB) have shown protective effects against DKD in experimental and clinical trials [1]. Angiotensin II is known to be produced from angiotensin I by ACE in general, but chymase can also generate angiotensin II from angiotensin I in cardiovascular tissues [2,3]. Chymase inhibitors have been reported to be useful against cardiovascular diseases such as cardiac dysfunction and atherosclerosis [4,5]. Both ACE inhibitors and ARBs are known not only to reduce blood pressure but also to increase plasma renin activity, while chymase inhibitors do not affect blood pressure and plasma renin activity [6]. These findings suggest that chymase inhibitors, but not ACE inhibitors and ARBs, are uninvolved in the regulation of systemic angiotensin II levels. Chymase is a chymotrypsin-like serine protease, and its activity is rapidly abolished by endogenous serine protease inhibitors in the blood [7]. Therefore, chymase-dependent angiotensin II formation is observed in local tissues, but not systemically [8]. Angiotensin II plays a crucial role in podocyte injury; high glucose increases angiotensin II formation in cultured mouse podocytes, and its increase can be inhibited by a chymase inhibitor but not by an ACE inhibitor [9]. Furthermore, a significant increase in glomerular chymase expression is observed in patients with diabetic nephropathy [10]. These reports suggest that chymase-dependent angiotensin II formation may play an important role in the pathogenesis of DKD, especially podocyte injury.

Chymase is also able to produce an active form of transforming growth factor (TGF)-β from the latent TGF-β-binding protein in cultured fibroblasts [11]. TGF-β induces renal fibrosis, resulting in the progression of DKD [12]. An augmentation of TGF-β gene expression is observed along with progressive podocyte injury in mice [13]. Both angiotensin II and high glucose induce TGF-β gene expression in kidneys, resulting in an increase of albuminuria [14]. Previously, we demonstrated that both angiotensin II and TGF-β in the liver were significantly increased in non-alcoholic steatohepatitis animal models, but they were significantly reduced by treatment with a chymase inhibitor [15,16]. Therefore, chymase may promote podocyte injury via upregulation of TGF-β in addition to angiotensin II in DKD. However, there have been few reports showing a relationship between chymase and podocyte injury in type 2 diabetes, and the mechanism remains unclear [17,18].

The db/db mouse is a genetic obese diabetic model that develops abnormalities in renal function and morphology that resemble those in patients with type 2 diabetes mellitus. In the present study, we evaluated the effect of a specific chymase inhibitor on albuminuria, which is an indicator of podocyte injury, in db/db mice and clarified its mechanism.

## 2. Results

### 2.1. Body Weight, Fasting Blood Glucose Level, and Urinary Albumin/Creatinine Ratio

At 6 weeks old, body weights in both diabetic groups of db/db mice before treatment with placebo and TY-51469 were significantly heavier than in the normal group of db/m mice (Figure 1a). The significant differences between normal and placebo-treated groups and between normal and TY-51469-treated groups continued until 4 weeks after starting treatment, and no significant difference between placebo- and TY-51469-treated groups was observed (Figure 1a). 

Fasting blood glucose levels were significantly higher in the diabetic groups before treatment with placebo and TY-51469 than in the normal group, and no significant difference between placebo- and TY-51469-treated groups was observed throughout the experimental period (Figure 1b).

Significant augmentations of urinary albumin/creatinine ratio were observed in the diabetic groups before treatment with placebo or TY-51469 compared with the normal group (Figure 1c). However, unlike body weight and fasting blood glucose level, albumin/creatinine ratios were significantly reduced by treatment with TY-51469 at 2 and 4 weeks after starting treatment (Figure 1c).

### 2.2. Renal mRNA Level and Activity of Chymase

The renal mRNA level of mouse mast cell protease (MMCP-4), which is a mouse chymase, was significantly higher in the placebo-treated group than in the normal group, but it was significantly lower in the TY-51469-treated group than in the placebo-treated group (Figure 2a).

Renal chymase activity was significantly increased in the placebo-treated group compared with the normal group, but it was reduced by treatment with TY-51469 (Figure 2b).

### 2.3. NADPH Oxidase (NOX)4 mRNA Level, and Malondialdehyde Level in Kidneys

The renal NOX4 mRNA level was significantly augmented in the placebo-treated group compared with the normal group, but it was significantly attenuated in the TY-51469-treated group (Figure 3a). A significant augmentation of the oxidative marker malondialdehyde in kidneys was also observed in the placebo-treated group, but it was significantly attenuated by treatment with TY-51469 (Figure 3b).

### 2.4. Renal mRNA Levels of Tumor Necrosis Factor (TNF)-α and TGF-β

Significant increases of TNF-α and TGF-β mRNA levels in kidneys were observed in the placebo-treated group compared with the normal group, but they were significantly reduced by treatment with TY-51469 (Figure 4a,b). 

### 2.5. Linear Regression Analyses of Renal mRNA Levels

A significant correlation between MMCP-4 and NOX4 mRNA levels was observed (Figure 5a). Relationships between MMCP-4 and TNF-α and between MMCP-4 and TGF-β were also significantly correlated (Figure 5b,c). 

### 2.6. Histological Analysis of Glomeruli

Representative images of glomeruli stained with periodic acid-Schiff (PAS) staining in normal, placebo-, and TY-51469-treated mice are shown in Figure 6a. No glomerulus histological changes were observed in any group.

Representative images of glomeruli stained with immunohistochemical staining of nephrin are shown in Figure 6a. Nephrin is a specific podocyte protein which reduction indicates podocyte injury. The nephrin-positive area was significantly smaller in the placebo-treated group than in the normal group, but it was significantly improved by treatment with TY-51469 (Figure 6b).

### 2.7. Numbers of Mast Cells and Chymase-Positive Cells in Kidneys

Representative images of toluidine blue-stained cells as mast cells in kidney sections from normal, placebo-, and TY-51469-treated mice are shown in Figure 7a. In the placebo-treated group, a significant increase of mast cell number was observed compared with the normal group, but it tended to be decreased by treatment with TY-51469 (Figure 7b).

Representative images of MMCP-4-positive cells in kidney sections from normal, placebo-, and TY-51469-treated mice are shown in Figure 7a. The MMCP-4 positive cell number in the placebo-treated group was significantly increased compared with the normal group, but it was significantly reduced in the TY-51469-treated group (Figure 7c).

## 3. Discussion

In the present study, body weight, fasting blood glucose levels and urinary albumin/creatinine ratio had already been higher in diabetic db/db mice than in normal db/mice before the start of the experiment. All of them were further increased in the placebo-treated db/db mice compared with normal db/m mice 4 weeks after the start of the experiment. In the placebo-treated db/db mice, renal chymase activity was significantly higher in the normal db/m mice, but it was significantly reduced by treatment with a chymase inhibitor TY-51469. The chymase inhibitor-treated db/db mice significantly lowered the urinary albumin/creatinine ratio compared with the placebo-treated db/db mice. However, chymase inhibition did not affect body weight or fasting blood glucose levels, both of which were augmented in db/db mice. Mouse chymase, MMCP-4, is able to produce angiotensin II from angiotensin I [19]. In immune complex-mediated glomerulonephritis, both significant reductions of glomerular angiotensin II and urinary protein have been reported in MMCP-4-deficient mice [20]. In cultured mouse podocytes, high glucose increases the angiotensin I concentration via upregulation of renin, which also upregulates angiotensin II; this is significantly reduced by a chymase inhibitor but not by an ACE inhibitor [9]. In a previous report, it was found that an ARB reduces urinary albumin excretion without reducing body weight and blood glucose levels in db/db mice [21]. Therefore, upregulated chymase may play a crucial role in the augmentation of urinary albumin excretion via upregulation of renal angiotensin II formation in db/db mice. In fact, we found a significant reduction of renal chymase activity using a specific chymase inhibitor, which prevented the augmentation of urinary albumin excretion without reducing hyperglycemia in db/db mice.

Podocytes are highly specialized and differentiated epithelial cells and important structural elements of the glomerular capillary filtration barrier. Studies have shown that glomerular dysfunction causing proteinuria is typically associated with foot process effacement and slit diagram disruption [22,23]. Nephrin is a glomerular epithelial glycoprotein that plays a crucial role in the structure of the filtration diaphragm [24]. A progressive decline in the expression of nephrin indicates damage to the cytoskeleton and filtration diaphragm in podocytes, which leads to proteinuria. We demonstrated for the first time that chymase inhibition attenuated the disappearance of nephrin in db/db mice. An ARB was previously reported to attenuate the effect of nephrin downregulation, resulting in urinary albumin excretion in streptozotocin-induced diabetic rats [25]. Therefore, the mechanism of chymase inhibition to prevent podocyte injury may be dependent on the suppression of chymase-dependent angiotensin II formation in the glomerulus.

The renal malondialdehide level was increased in db/db mice treated with placebo in this study. Malondialdehide is derived from lipid hydroperoxides produced by oxidative stress, and its level is used as an indicator of oxidative stress. Clinical and experimental studies have shown an important role for oxidative stress in diabetic nephropathy [26,27]. Chymase inhibition resulted in a reduction of renal malondialdehide levels, demonstrating that chymase induces oxidative stress in the diabetic model. Angiotensin II is a key factor that induces oxidative stress via an NADPH oxidase NOX4 upregulation in glomeruli and promotes podocyte dysfunction in the development of DKD [28]. Podocyte-specific NOX4-deficient mice show a reduction of oxidative stress and attenuation of nephrin disappearance, resulting in the reduction of urinary albumin excretion in streptozotocin-induced diabetic mice [29]. In this study, the disappearance of glomerular nephrin expression was significantly recovered by chymase inhibition, and the augmentation of NOX4 mRNA levels was also reduced by chymase inhibition. Furthermore, a significant correlation between chymase (MMCP-4) and NOX4 mRNA levels was also observed. The augmentation of chymase-dependent angiotensin II formation may induce podocyte injury via upregulation of NOX4-dependent oxidative stress.

We observed that the upregulation of TNF-α mRNA levels in db/db mice was significantly reduced by chymase inhibition, and there was a significant correlation between chymase (MMCP-4) and TNF-α mRNA levels. Previous studies have reported that renal inflammation promotes the development and progression of DKD, and TNF-α is a major inflammatory cytokine that plays an important role in the pathogenesis of DKD including podocyte injury [30,31]. Urinary TNF-α excretion is elevated in type 2 diabetic patients as compared with nondiabetic individuals, and it correlates with urinary albumin excretion [32]. An anti-TNF-α monoclonal antibody attenuates urinary albumin excretion in streptozotocin-induced diabetic rats [33]. Angiotensin II is known to increase the gene expression of inflammatory cytokines such as TNF-α, and an ARB has been used to attenuate both the augmented renal gene expression of TNF-α and podocyte injury in db/db mice [21]. The reduction of TNF-α mRNA levels by chymase inhibition may depend on chymase-dependent angiotensin II reduction, and this mechanism may contribute to prevent podocyte injury. However, TNF-α is able to induce the activation of NADPH oxidase in isolated rat glomeruli, and this augmentation of oxidative stress results in alterations of glomerular dysfunction and consequently increased albumin permeability [34,35]. Therefore, the mechanism by which chymase inhibition prevents podocyte injury may partly include the reduction of oxidative stress directly promoted by TNF-α.

It has been reported that renal TGF-β mRNA is induced by both angiotensin II and hyperglycemia [14]. In a type 1 diabetic mouse model, the expression of nephrin decreases and urinary albumin excretion increases as the expression of TGF-β increases [13]. In immortalized human podocytes, TGF-β induces dynamic changes in morphology that appear as retractions and shortening of the foot processes, and then broad and complex tight junctions are formed between adjacent podocytes [36]. This differentiation is associated with a reduction of glomerular epithelial proteins such as nephrin [12]. We observed the augmentation of TGF-β mRNA levels in db/db mice as previously reported [21], and the upregulation of TGF-β mRNA levels was significantly attenuated by chymase inhibition. We also observed a significant correlation between chymase (MMCP-4) and TGF-β mRNA levels. Chymase inhibition was not affected by the hyperglycemia in db/db mice, but chymase-dependent angiotensin II may cause the augmentation of TGF-β mRNA levels. On the other hand, chymase can activate not only angiotensin II but also TGF-β from their precursors [11]. TGF-β is known to promote the proliferation of fibroblasts. In human cultured fibroblasts, chymase increased TGF-β concentration in the medium several minutes after the injection of chymase and dose-dependently induced the cell proliferation [11]. This proliferation was completely suppressed by anti-TGF-β neutralizing antibodies or a chymase inhibitor, but not by an ARB [11]. This indicates that the activation of TGF-β from latent TGF-β-binding protein in the fibroblasts by chymase caused the proliferation of fibroblasts. Therefore, chymase inhibition may contribute to the amelioration of TGF-β-induced podocyte injury via the suppression of TGF-β activation in addition to the reduction of TGF-β gene expression.

Both mast cells and MMCP-4-positive cells were significantly increased in number in placebo-treated db/db mice in the present study. Increases in both mast cells and chymase-positive cells have been reported in human diabetic nephropathy [10,37]. We also observed that the specific chymase inhibitor TY-51469 showed a reduction of mast cell and MMCP-4-positive cell numbers. Numerous studies have demonstrated that chymase inhibitors can reduce mast cell numbers in various experimental models [38,39,40,41]. Chymase is known to enzymatically cleave the inactive membrane-bound form of stem cell factor (SCF) to the active form of SCF, and the activated SCF stimulates the development and proliferation of mast cells, resulting in increased mast cell numbers [42]. In the present study, the chymase inhibitor reduced not only chymase activity but also chymase mRNA levels, indicating that chymase inhibition may reduce chymase activity indirectly via a reduction of chymase-generating mast cell numbers in addition to directly inhibiting chymase activity.

One limitation of this study is that no morphological change of glomeruli was observed. The reason may be the early stage of DKD, because we evaluated the db/db mice at 10 weeks old. In a previous report, abnormalities in renal morphology such as expansion of mesangium were remarkable in 18-week-old db/db mice [21]. In the future, whether chymase inhibition is useful for improving chronic diabetic nephropathy should be evaluated.

## 4. Materials and Methods

### 4.1. Drug

TY-51469 was synthesized as a specific chymase inhibitor and obtained from Toaeiyo Ltd. (Tokyo, Japan). TY-51469 inhibits chymase activity with an IC50 of 0.7 nM, but it does not inhibit ACE activity [43]. The chemical structure of the chymase inhibitor TY-51469 is shown in Figure 8.

### 4.2. Animal and Experimental Design

All animal procedures were approved by the Committee of Animal Use and Care of Osaka Medical College (Approval code number: 2019-019, 17 May 2019) and performed in accordance with the Guidelines for Animal Research. Five-week-old male db/db mice and their nondiabetic db/m littermates were purchased from Japan SLC (Shizuoka, Japan). The animals were maintained 12-h light/12-h dark conditions. Temperature and humidity were controlled, and animals had free access to water and food.

At 6 weeks old, db/db mice were equally divided into two groups based on body weight, blood glucose level after a 16-h fasting period, and urinary albumin/creatinine ratio. Blood glucose level was measured by Accu-Chek Performa glucose test strips (Roche Diagnostics, Tokyo, Japan). Urinary albumin and creatinine levels were measured by SRL Inc. (Tokyo, Japan). The diabetic mice were treated concurrently with either TY-51469 (10 mg/kg/day, *n* = 10) or placebo (saline, *n* = 10) for 4 weeks. The drugs were administered subcutaneously using an Alzet osmotic minipump (model 2004; Durect, Cupertino, CA, USA). Nondiabetic db/m mice were used as a normal group (*n* = 6).

### 4.3. Chymase Activity and Malondialdehyde Level in Kidneys

Kidneys were homogenized in 20 mM sodium phosphate buffer, pH 7.4. The homogenate was centrifuged at 15,000 *g* for 30 min. The supernatant was discarded, and the pellet was resuspended and homogenized in 10 mM sodium phosphate buffer, pH 7.4, containing 2 M KCl and 0.1% Nonidet P-40. The homogenate was centrifuged at 15,000 *g* for 30 min, and the supernatant was used for measurements of enzyme activity and malondialdehyde level [44].

Chymase activity was measured using a synthetic substrate, Suc-Ala-Ala-Pro-Phe-4- methylcoumaryl-7-amide, specifically designed as a substrate for chymase (Peptide Institute Inc., Osaka, Japan) [45]. One unit of chymase activity was defined as the amount of enzyme required to cleave 1 μmol of 7-amino-4-methyl-coumarin/min.

The measurement of malondialdehyde, a product of lipid peroxidation, was conducted using a commercial kit (Cayman Chemical Company, Ann Arbor, MI, USA), which is based on the principle of quantifying the product from the reaction of malondialdehyde and thiobarbituric acid according to the manufacturer’s instructions [46]. The protein concentration was assayed using BCA Protein Assay Reagents (Pierce, Rockford, IL, USA), with bovine serum albumin as the standard.

### 4.4. Real-Time Polymerase Chain Reaction (RT-PCR)

Renal total RNA was extracted using Trizol reagent (Life Technologies, Rockville, MD, USA) and subsequently dissolved in RNase-free water (Takara Bio Inc., Otsu, Japan) [47]. Total RNA (1 μg) was transcribed into cDNA with Superscript VIRO (Invitrogen, Carlsbad, CA, USA). Then, mRNA levels were measured by RT-PCR on a Stratagene Mx3000P (Agilent Technologies, San Francisco, CA, USA) using TaqMan fluorogenic probes. RT-PCR primers and probes for MMCP-4 and NOX4, TNF-α, TGF-β, and 18S ribosomal RNA (rRNA) were designed by Roche Diagnostics (Tokyo, Japan). The primers were as follows: 5’-ggcctgtaaaaactattggcatt-3’ (forward) and 5-cacacagtagaggtcctccaga-3’ (reverse) for MMCP-4, 5’-ccctaaacgttctacttttctgga-3’ (forward) and 5-tgctctgcttaaacacaatcct-3’ (reverse) for NOX4, 5’-aggcgaagattactgccaag-3’ (forward) and 5’-catggctatgaggtagagacagg-3’ (reverse) for TNF-α, 5’-tggagcaacatgtggaactc-3’ (forward) and 5’-cagcagccggttaccaag-3’ (reverse) for TGF-β, and 5’-gcaattattccccatgaacg-3’ (forward) and 5’-gggacttaatcaacgcaagc-3’ (reverse) for 18S rRNA. The probes were as follows: 5’-tccaggtc-3’ for MMCP-4, 5’- tcctgctg-3’ for NOX4, 5’-agccccag-3’ for TNF-α, 5’-ttcctggc-3’ for TGF-β, and 5’-ttcccagt-3’ for 18S rRNA. The mRNA levels of MMCP-4, NOX4, TNF-α, and TGF-β were normalized to that of 18S rRNA.

### 4.5. Histological Analysis

Renal tissue specimens were fixed with Carnoy’s fixative in 10% methanol overnight. Fixed renal tissues were embedded in paraffin, and then cut at a thickness of 4 μm. The sections were mounted on adhesive glass slides (Matsunami Glass Ind., Kishiwada, Japan) and deparaffinized with lemosol (Wako Pure Chemicals, Osaka, Japan). The glomerulus histological change was assessed using PAS staining [48]. Mast cells were stained with 0.05% toluidine blue (Chroma-Gesellschaft, Stuttgart, Germany) [47].

Immunohistological examinations were performed using an anti-nephrin antibody (LS-B1382, LSBio, Seattle, WA, USA) and anti-MMCP-4 antibody (ab92368, Abcam, Cambridge, UK) according to a protocol described elsewhere [48]. In brief, to suppress endogenous peroxidase activity and nonspecific binding, the deparaffinized sections were incubated with 3% hydrogen peroxide and protein-blocking solution for 5 min at room temperature, respectively. Then, these sections were incubated with the above diluted primary antibodies overnight at 4 °C, followed by reaction with components from a labeled streptavidin-biotin peroxidase kit (Dako LSAB kit, Dako, Carpinteria, CA, USA) that included 3-amino-9-ethylcarbazole color development. Sections were then lightly counterstained with hematoxylin.

### 4.6. Statistical Analysis

All numerical data are expressed as means ± standard error of the mean (SEM) and were analyzed with BellCurve® statistical analysis software for Microsoft Excel® (Tokyo, Japan). Significant differences among the mean values of multiple groups were evaluated by one-way analysis of variance (ANOVA) followed by a post hoc analysis (Fisher’s test). Pearson’s correlation coefficient was measured to test the linear relationship between two variables using linear regression analysis. *P* < 0.05 was considered significant.

## 5. Conclusions

We demonstrated that the mechanism by which a chymase inhibitor reduced urinary albumin excretion was via the prevention of podocyte injury by oxidative stress.

## Figures and Tables

**Figure 1 ijms-21-07495-f001:**
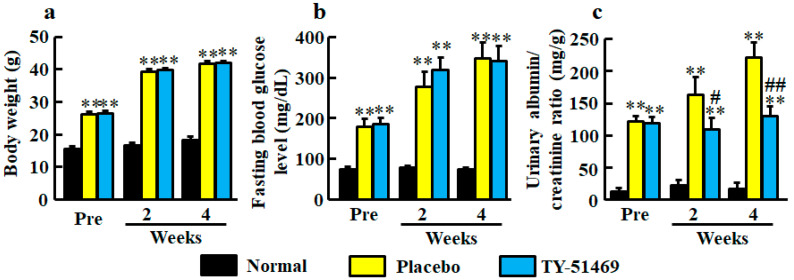
Body weight (**a**), fasting blood glucose level (**b**), and urinary albumin/creatinine ratio (**c**) in normal, placebo-, and TY-51469-treated groups before (Pre) and 2 and 4 weeks after starting treatment. Values represent mean ± SEM. ** *P* < 0.01 vs. normal group. ^#^
*P* < 0.05 and ^##^
*P* < 0.01 vs. placebo-treated group.

**Figure 2 ijms-21-07495-f002:**
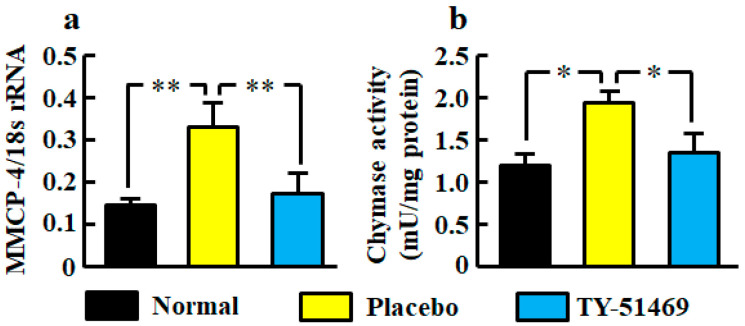
Renal MMCP-4 mRNA level (**a**) and chymase activity (**b**) in normal, placebo-, and TY-51469-treated groups 4 weeks after starting treatment. Values represent mean ± SEM. * *p* < 0.05 and ** *p* < 0.01 vs. placebo-treated group.

**Figure 3 ijms-21-07495-f003:**
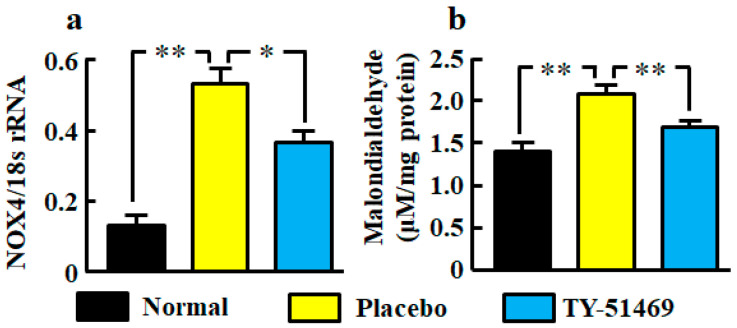
Oxidative stress markers NOX4 mRNA (**a**) and malondialdehyde (**b**) levels in kidneys from normal, placebo-, and TY-51469-treated mice 4 weeks after starting treatment. Values represent mean ± SEM. * *p* < 0.05 and ** *p* < 0.01 vs. placebo-treated group.

**Figure 4 ijms-21-07495-f004:**
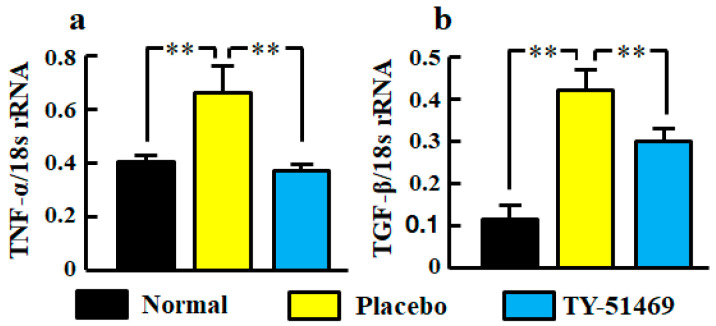
Renal mRNA levels of TNF-α (**a**) and TGF-β (**b**) in normal, placebo-, and TY-51469-treated groups 4 weeks after starting treatment. Values represent mean ± SEM. ** *p* < 0.01 vs. placebo-treated group.

**Figure 5 ijms-21-07495-f005:**
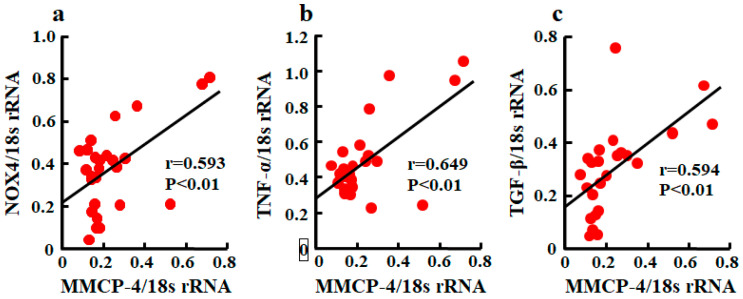
Linear regression analyses of correlations between MMCP-4 and NOX4 mRNA levels (**a**), between MMCP-4 and TNF-α mRNA levels (**b**), and between MMCP-4 and TGF-β mRNA levels (**c**) in kidneys of mice 4 weeks after starting treatment. Significant correlations were observed for all three.

**Figure 6 ijms-21-07495-f006:**
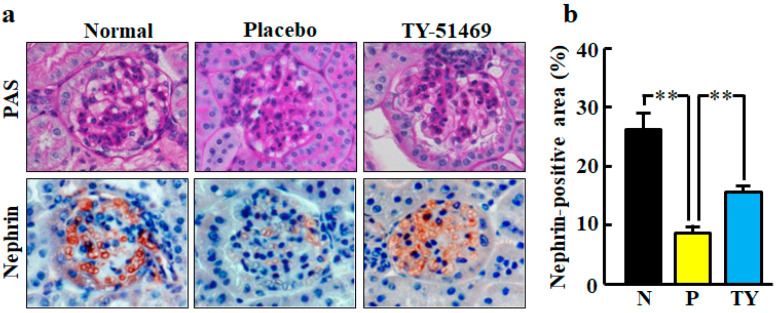
Representative images of glomeruli stained with PAS and immunostained with anti-nephrin antibody (nephrin-positive cells) from normal, placebo-, and TY-51469-treated mice 4 weeks after starting treatment (**a**). Original magnification was 200× (**a**). Ratio of nephrin-positive area to total glomerular area in normal (N), placebo (P)-, and TY-51469 (TY)-treated mice 4 weeks after starting treatment (**b**). Values represent mean ± SEM. ** *p* < 0.01 vs. placebo-treated group.

**Figure 7 ijms-21-07495-f007:**
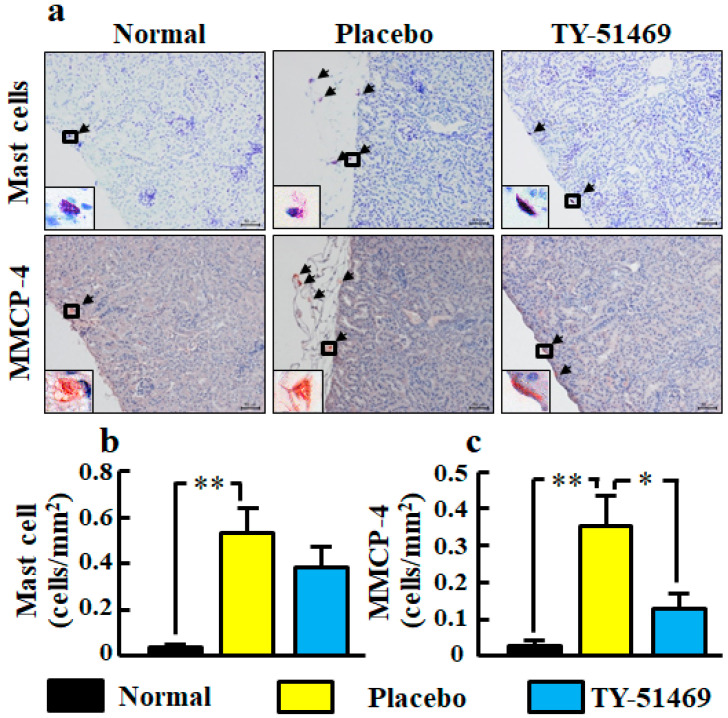
Representative images of kidney sections stained with toluidine blue (mast cells) and immunostained with anti-MMCP-4 antibody (MMCP-4-positive cells) from normal, placebo-, and TY-51469-treated mice 4 weeks after starting treatment (**a**). Black arrows indicate mast cells and MMCP-4-positive cells (**a**). The black frame is enlarged to the lower left corner (**a**). Original magnification was 200× (**a**). Numbers of mast cells (**b**) and MMCP-4-positive cells (**c**) in kidney sections in normal, placebo-, and TY-51469-treated mice 4 weeks after starting treatment. Values represent mean ± SEM. * *p* < 0.05 and ** *p* < 0.01 vs. placebo-treated group.

**Figure 8 ijms-21-07495-f008:**
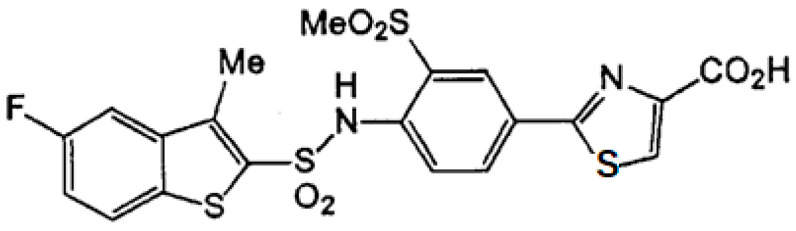
The chemical structure of TY-51469.

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
