# Peer review of "Mechanism of Albuminuria Reduction by Chymase Inhibition in Diabetic Mice"

_ijms, 2020, doi:10.3390/ijms21207495_

Round 1

Reviewer 1 Report

The paper by professor Takai and colleagues reports a study on the mechanism of albuminuria reduction by chymase inhibition in diabetic mice. TY-51469 was used as the chymase inhibitor. A number of parameters, including body weight, fasting blood glucose level, urinary albumin/creatinine ratio, renal mRNA level, NADPH oxidase NOX4 mRNA level, malondialdehyde level in kidneys, TNF-α and TGF-β mRNA levels were investigated for TY-51469-treated and placebo-treated groups of diabetic mice. Histological analysis of glomeruli was also performed. The study demonstrated that the use of chymase inhibitor significantly reduced the albumin/creatinin ratio. Additionally, NOX4 expression and the concentration of the oxidative stress marker malondialdehyde resulted decreased in the TY-51469-treated group. TNF-α and TGF-β mRNA levels were also reduced in the TY-51469-treated group. On the basis of the obtained results, the authors concluded that the mechanism by which the chymase inhibitor reduces urinary albumin excretion is through the prevention of podocyte damage by oxidative stress. 

In my opinion, the paper reports interesting results and describes a well-conducted research. Therefore, I recommend the publication of the manuscript in Int. J. Mol. Sci

For sake of clarity, I would only suggest to report the chemical structure of the chymase inhibitor TY-51469 used in the present study. 

Author Response

Thank you for your valuable suggestions.

We added Figure 8 as the chemical structure of TY-51469 to the revised manuscript (lines 256-258)

The chemical structure of the chymase inhibitor TY-51469 is shown in Figure 8.

Reviewer 2 Report

This is an interesting manuscript about inhibitor TY-51469 activity on DKD in diabetic mice. It is well written and logically described.

In line 58 different fonts were used in the same sentence, but the manuscript is overall nicely written and results are presented in a detailed and informative way.

The only recommendation is to discuss results that were significantly different in normal and placebo mice more thoroughly, and to end the discussion with the better explanation of mice age and its connection to the future experiments considering chronic diabetic nephropathy.

Author Response

Thank you for your valuable suggestions.

We corrected the fonts in line 58.

However, there have been few reports showing a relationship between chymase and podocyte injury in type 2 diabetes, and the mechanism remains unclear [17,18].

In the first paragraph of the discussion, we added the results of the differences between normal db/m mice and placebo-treated db/db mice.

(Lines 152-160)

In the present study, body weight, fasting blood glucose levels and urinary albumin/creatinine ratio had already been higher in diabetic db/db mice than in normal db/mice before the start of the experiment. All of them were further increased in the placebo-treated db/db mice compared with normal db/m mice 4 weeks after the start of the experiment. In the placebo-treated db/db mice, renal chymase activity was significantly higher in the normal db/m mice, but it was significantly reduced by treatment with a chymase inhibitor TY-51469. The chymase inhibitor-treated db/db mice significantly lowered the urinary albumin/creatinine ratio compared with the placebo-treated db/db mice. However, chymase inhibition did not affect body weight or fasting blood glucose levels, both of which were augmented in db/db mice.

We added the discussion with the better explanation of mice age and its connection to the future experiments considering chronic diabetic nephropathy.

(Lines 247-251)

One limitation of this study is that no morphological change of glomeruli was observed. The reason may be the early stage of DKD, because we evaluated the db/db mice at 10 weeks old. In a previous report, abnormalities in renal morphology such as expansion of mesangium were remarkable in 18-week-old db/db mice [21]. In the future, whether chymase inhibition is useful for improving chronic diabetic nephropathy should be evaluated.